# Pseudocapacitive Behavior of Blade-Coated Mo_1.33_CT_x_ i-MXene Electrodes in Aqueous Electrolytes

**DOI:** 10.3390/nano15201593

**Published:** 2025-10-19

**Authors:** Alexey Tsyganov, Olga Grapenko, Evgeniy Korotaev, Alexander Shindrov, Andrei Alferov, Alexander Gorokhovsky, Nikolay Gorshkov

**Affiliations:** 1Department of Chemistry and Technology of Materials, Yuri Gagarin State Technical University of Saratov, 77 Polytecnicheskaya Street, 410054 Saratov, Russia; andrey_080202@mail.ru (A.A.); algo54@mail.ru (A.G.); 2Research Institute of Physics, Southern Federal University, 194 Stachki Avenue, 344011 Rostov-on-Don, Russia; grapenko@sfedu.ru; 3Nikolaev Institute of Inorganic Chemistry, Siberian Branch, Russian Academy of Sciences, 630090 Novosibirsk, Russia; korotaev@niic.nsc.ru; 4Institute of Solid State Chemistry and Mechanochemistry, Siberian Branch of the Russian Academy of Sciences, 18 Kutateladze, 630090 Novosibirsk, Russia; a.shindrov@yandex.ru

**Keywords:** MXene, molybdenum carbide, nanosheet, capacitance, energy storage, supercapacitor

## Abstract

Two-dimensional molybdenum carbide (Mo_1.33_CT_x_ MXene) with ordered vacancies is one of the most promising materials for electrochemical energy storage. However, the high defectivity and tendency to aggregate of nanosheets hinders the large-scale fabrication of highly efficient Mo_1.33_CT_x_ -based electrodes. In this study, Mo_1.33_CT_x_/carbon nanotubes (CNTs) electrodes of varying thicknesses were fabricated using a scalable doctor blade technique. Their electrochemical performance was studied in H_2_SO_4_, H_3_PO_4_, LiCl and KCl electrolytes using cyclic voltammetry and galvanostatic charge–discharge methods. Electrodes with an active material mass loading of 1.6 mg/cm^2^ exhibited specific capacitances of 352, 287, 172, and 107 F/g in H_2_SO_4_, H_3_PO_4_, LiCl, and KCl electrolytes, respectively, at a scan rate of 2 mV/s. Increasing the mass loading of the electrode material to 3.5 mg/cm^2^ resulted in a specific capacitance of 349, 260, 162 and 98 F/g in the same electrolytes. The incorporation of CNTs enabled rapid electrolyte ion transport throughout the electrode bulk, maintaining high capacitance values even at high scan rates. These results open new avenues for the development of high-performance electrode materials for supercapacitors.

## 1. Introduction

The ongoing pursuit of efficient and sustainable energy storage systems has spurred significant research into novel materials capable of addressing the growing demands of modern energy technologies. Among various electrochemical energy storage systems, pseudocapacitors have emerged as a promising class of devices that combine high power density with excellent cycling stability [1,2,3,4]. Within this class, two primary types are distinguished based on their charge storage mechanism: electric double-layer capacitors (EDLC), where energy storage occurs via the electrostatic adsorption of ions at the electrode-electrolyte interface [5], and pseudocapacitors, where fast and reversible Faradaic (redox) reactions on the electrode surface are added to the electrostatic storage [6]. It is the pseudocapacitive materials that enable a substantial increase in the specific capacitance of the device compared to purely double-layer systems. These features make them highly suitable for applications ranging from grid stabilization and electric vehicles to portable electronics and wearable technologies [7]. Since the performance of pseudocapacitive supercapacitors is largely determined by the properties of the electrode materials, which must support fast redox reactions, high electrical conductivity, and high chemical stability across various electrolytes [8,9], the development of novel electrodes for them is of paramount importance and represents a high-priority research area. A key research direction involves the design of electrode materials that integrate high electrical conductivity, long-term cycling stability, and the high electrochemical activity of pseudocapacitive components. This approach paves the way for devices achieving battery-level energy density without compromising their high power output and long-term durability.

MXenes, a family of two-dimensional transition metal carbides and nitrides, have gained considerable attention in recent years as a new generation of nanomaterials for electrochemical energy storage [10,11,12,13,14]. MXenes are typically produced by selective etching of A elements from nanolaminar M_n+1_AX_n_ (MAX) phases (where M is an early transition metal, A is a group IIIA or IVA element, and X is carbon and/or nitrogen), resulting in two-dimensional structures with a general formula of M_n+1_X_n_T_x_ [15,16,17]. Here, T represents surface terminations such as –OH, –F, or –O, introduced during the etching process [18]. The unique combination of metallic conductivity, hydrophilic surfaces, and structural tunability endows MXenes with remarkable electrochemical properties, including high capacitance, fast ion transport, and accessible surface redox activity [19,20,21,22,23,24,25].

Among the various MXene compositions, Mo_1.33_CT_x_ MXene with ordered vacancies has attracted particular attention in the field of electrochemical energy storage [26,27]. This MXene is derived from the in-plane chemically ordered (Mo_2/3_Re_1/3_)_2_AlC i-MAX phase (Re—rare earth elements) [28,29,30]. A notable feature of i-MAX phases is that Re atoms can be readily removed together with the Al atomic layer during the etching process [31]. As a result, the obtained Mo_1.33_CT_x_ MXene exhibits periodic vacancies at Re atomic sites. These vacancies lead to modifications in the electronic structure, enhanced surface reactivity, and improved ion accessibility, which are especially beneficial for supercapacitor applications. Furthermore, Mo_1.33_CT_x_ demonstrates robust electrochemical stability in both acidic and neutral aqueous electrolytes [32,33]. Despite these advantages, a significant challenge in the practical application of Mo_1.33_CT_x_ lies in its tendency to restack during electrode fabrication. This restacking, driven by van der Waals forces and capillary effects during solvent evaporation, severely limits electrolyte ion diffusion between layers and reduces the accessible surface area for charge storage [34]. Consequently, the full electrochemical potential of MXenes cannot be realized in densely packed films. Moreover, unlike other MXenes, Mo_1.33_CT_x_ does not form flexible electrode films due to its defective structure. This imposes serious limitations on the scalable fabrication of high-performance Mo_1.33_CT_x_-based electrodes. Therefore, to enable large-scale manufacturing of such electrodes, it is necessary to introduce additional binders and spacers to prevent nanosheet aggregation. This approach enables the fabrication of electrodes that are mechanically robust and provide superior electrolyte ion transport throughout their bulk. One effective strategy to mitigate restacking is the incorporation of carbon nanotubes (CNTs) into the MXene matrix. CNTs act as spacers that prevent dense re-aggregation of nanosheets, while also enhancing the electronic conductivity and mechanical stability of the composite film. Several studies have demonstrated that CNTs improve ion accessibility, increase charge storage capacity, and enable flexible, free-standing electrodes based on MXenes [35,36,37].

To enable scalable production of high-performance MXene-based electrodes, it is essential to employ fabrication methods that ensure uniformity, controllability, and reproducibility. This is particularly relevant for Mo_1.33_CT_x_ MXene, as the fabrication of its freestanding films is challenging due to their tendency to crack. Moreover, it should be emphasized that systematic studies on the fabrication and optimization of electrodes specifically based on Mo_1.33_CT_x_ have not been conducted hitherto. In this context, the blade-coating method (also known as “doctor blade”) proves to be a critically important technology and is a primary candidate for creating high-performance electrodes. Its advantages—simplicity, cost-effectiveness, and suitability for large-area processing—allow it to circumvent the problem of the material’s brittleness by enabling coating directly onto current collectors. The most significant advantage of the blade-coating method is the ability to precisely control the film thickness and morphology, which directly dictates the trade-off between areal capacitance and rate performance in practical devices [38,39,40,41]. The successful application of this technique in battery and printed electronics manufacturing makes it the most preferable and promising candidate for addressing the task at hand. Accordingly, it is important to optimize the scalable fabrication of Mo_1.33_CT_x_ MXene electrodes for their application in supercapacitors.

Herein, we investigate the electrochemical energy storage properties of Mo_1.33_CT_x_ MXene electrodes fabricated via a scalable blade-coating process. Mo_1.33_CT_x_ MXene was obtained by selective etching of the (Mo_2/3_Re_1/3_)_2_AlC MAX phase under hydrothermal conditions, which provided a mild environment for the extraction of Al and Y atoms while preserving the integrity of the two-dimensional structure. This hydrothermal synthesis yields high-quality, delaminated monolayers suitable for further electrode processing. To suppress restacking and enhance conductivity, the delaminated Mo_1.33_CT_x_ nanosheets were mixed with carbon nanotubes (CNTs), forming a composite suspension. Polyvinylidene fluoride (PVDF) was added as a binder to improve adhesion to the current collector. The resulting slurry was deposited onto metal foil substrates using the blade-coating method, ensuring controlled film thickness and uniformity over large areas. This scalable and reproducible approach represents a promising route for the industrial fabrication of MXene-based electrodes. The electrochemical performance of the blade-coated Mo_1.33_CT_x_–CNTs electrodes was investigated in a range of aqueous electrolytes, including H_3_PO_4_, H_2_SO_4_, LiCl, and KCl, to evaluate their behavior in different ionic environments. Additionally, the effect of electrode thickness on charge storage properties was assessed under various scan rates to elucidate diffusion dynamics and optimize electrode architecture for high-rate applications.

## 2. Materials and Methods

### 2.1. Synthesis of Mo_1.33_CT_x_ MXene Nanosheets

The quaternary (Mo_2/3_Y_1/3_)_2_AlC MAX phase was used to obtain the Mo_1.33_CT_x_ MXene. The (Mo_2/3_Y_1/3_)_2_AlC MAX phase powder was prepared via pressureless solid-state sintering. To achieve this, elemental powders of Mo (99.9%, 10 µm), Y (99.99%, 60 µm), Al (99.5%, 10–20 µm), and graphite (99.9%, 5 µm) were mixed in a molar ratio of 4:2:3.6:1. Excess aluminum was added to compensate for high-temperature volatilization. The powder mixture was homogenized using a Fritsch Pulverisette 6 planetary mill for 1 h. The homogenized mixture was then transferred into a graphite crucible and sintered in a tube furnace at 1500 °C for 2 h under a continuous argon atmosphere to prevent oxidation. The heating rate was maintained at 10 °C/min. After cooling, the sintered product was ground using a Fritsch Pulverisette 0 ball mill and sieved through a 400-mesh screen. To remove intermetallic impurities, the MAX phase powder was treated with 9 M HCl, followed by repeated washing with deionized water and drying at 60 °C. The resulting purified powder was used for MXene synthesis.

To obtain multilayer Mo_1.33_CT_x_ MXenes, a hydrothermal etching protocol of the (Mo_2/3_Y_1/3_)_2_AlC MAX phase in HCl-LiF mixture was used [42,43,44]. Specifically, 2.28 g of the MAX phase, 1.05 g of LiF, and 60 mL of 6 M HCl were combined in a 150 mL Teflon autoclave and heated at 170 °C for 24 h. This process selectively etched the weakly bonded Al and Y atoms from the nanolaminar MAX phase, resulting in a multilayered Mo_1.33_CT_x_ MXene structure. After cooling, the resulting dispersion was washed with deionized water to a pH of 6 using a vacuum filter. The resulting wet precipitate was then used to obtain single-layer Mo_1.33_CT_x_ flakes.

To obtain single-layer Mo_1.33_CT_x_ flakes, the multilayer MXene powder was intercalated with a 5 wt.% solution of tetramethylammonium hydroxide (TMAOH). For every 1 g of MXene, 5 mL of TMAOH solution were added, and the mixture was stirred vigorously for 1 h. After successful intercalation of TMA^+^ ions between the layers, the precipitate was washed with deionized water by centrifugation at 8000 rpm. It was then redispersed in deionized water (100 mL per gram of powder) and subjected to ultrasonication for 30 min at 10–15 °C. The formation of a stable colloidal solution indicated successful delamination. To remove undelaminated MXenes and any other impurities, the colloidal solution was centrifuged at 2000 rpm for 30 min. The supernatant contained predominantly single-layer Mo_1.33_CT_x_ flakes.

To further fabricate the electrode coatings, the delaminated MXene was transferred into dimethylformamide (DMF) via solvent exchange, without drying, as previously recommended [45]. The aqueous colloid was centrifuged at 8000 rpm for 1 h, and the sediment was redispersed in DMF. This process was repeated three times to remove the water. After the final centrifugation step, a 6 wt.% MXene slurry was obtained.

### 2.2. Preparation of Electrodes

MXene-based electrodes of varying thicknesses were fabricated using the doctor blade (blade coating) technique. A 6 wt.% Mo_1.33_CT_x_ MXene slurry in DMF was mixed with a 1 wt.% dispersion of carbon nanotubes (CNTs) and a 5 wt.% solution of polyvinylidene fluoride (PVDF) in DMF. The CNTs were incorporated to enhance the electrical conductivity of the resulting films. The mass ratio of Mo_1.33_CT_x_:CNTs:PVDF was maintained at 80:10:10. The resulting suspension was stirred continuously for 24 h to ensure uniform dispersion of all components. The homogeneous slurry was then cast onto metal foil substrates using an adjustable blade coater. After coating, the films were dried at 90 °C to remove the solvent. Adjusting the blade gap allowed for the preparation of electrodes with mass loadings of 1.6, 3.5, and 5.3 mg·cm^−2^.

### 2.3. Material Characterization

The phase composition of the synthesized materials was investigated using X-ray diffraction (XRD) on an ARL X’TRA diffractometer (Cu Kα radiation, λ = 0.15412 nm; Thermo Scientific, Ecublens, Switzerland). The morphology of the multilayered Mo1.33CTx MXene was examined by scanning electron microscopy (SEM) using a Hitachi S-3400N microscope. To analyze the morphology of single-layer MXene flakes, transmission electron microscopy (TEM) was conducted with a FEI Tecnai G2 Spirit Biotwin microscope (Netherlands). X-ray photoelectron spectroscopy (XPS) was performed on a SPECS spectrometer equipped with a monochromatic Al Kα X-ray source (hν = 1486.6 eV). All measurements were carried out at room temperature. The binding energy scale was calibrated using the C 1s peak of adventitious carbon at 284.7 eV (C–C component). XPS data analysis and peak fitting were conducted using CasaXPS software (Version 2.3.24PR1.0, Casa Software Ltd.; Teignmouth, UK). Molybdenum K-edge X-ray absorption near-edge structure (XANES) spectra were recorded using a laboratory-based Rigaku R-XAS spectrometer to evaluate the local electronic structure and oxidation state of molybdenum in the MXene samples.

### 2.4. Electrochemical Characterization

Electrochemical measurements were performed at 25 °C using a three-electrode configuration connected to a Smartsoft Stat PS-10-4 potentiostat/galvanostat (Elins, Chernogolovka, Russia). An Ag/AgCl electrode was used as the reference electrode and a graphite rod as the counter electrode. The working electrodes were metal substrates coated with Mo_1.33_CT_x_/CNTs/PVDF. Electrodes with an area of 1 cm^2^ and different coating thicknesses were used in the study. Electrochemical properties were studied using cyclic voltammetry (CV) at scan rates ranging from 2 to 100 mV·s^−1^ and galvanostatic charge–discharge (GCD). The potential window for different electrolytes was selected experimentally. These properties were studied in aqueous solutions of 2 M H_2_SO_4_ (pH ≈ 0), 1 M H_3_PO_4_ (pH ≈ 1.06), 5 M LiCl (pH = 7), and 1 M KCl (pH = 7).

The specific gravimetric capacitance was calculated from the cathodic portion of the CV curves using the formula for a three-electrode configuration.(1)$$C_{g}=\frac{\int IdU}{mv∆U},$$
where $C_{g}$ (F·g^−1^) is the specific gravimetric capacitance of the electrode, *v* is the scan rate, *m* is the working mass of the electrode, ∆*U* is the potential window, and ∫IdU is the integral area under the CV curve of the discharge current.

The specific capacitance was calculated using the GCD curves and the following equation:(2)$$C_{g}=j\int \frac{dt}{U(t)},$$
where *j* s is the constant current density, and *U(t)* is the working potential.

To assess the contribution of surface-controlled and diffusion-controlled processes, the kinetics of charge accumulation were studied. The current response on the CV curves at different scan rates can be expressed as follows:(3)$$i=av^{b},$$

This equation can be transformed into the following form:(4)$$\log\left(i\right)=\log\left(a\right)+blog(v),$$
here, *i* is the peak value of the current, *v* is the scan rate, *a* and *b* are parameters that can be determined from the $\log\left(i\right)$ versus log(v) plot. A dominance of surface-capacitive contributions is typical of *b* values close to 1, while a dominance of diffusion-controlled processes is typical of *b* values close to 0.5.

The following equation was used to quantify the contribution of surface-controlled and diffusion-controlled processes:(5)$$i=k_{1}v+k_{2}v^{1/2},$$
here, *i* is the current at a fixed potential, $k_{1}v$ is the surface-capacitive contribution, $k_{2}v^{1/2}$ is the contribution of diffusion-controlled processes. This equation can be converted into the following form:(6)$$i\cdot v^{-1/2}=k_{1}v^{1/2}+k_{2},$$

The coefficient $k_{1}$ was determined from the slope of the line on the graph showing the dependence $i\cdot v^{-1/2}$on $v^{1/2}$.

The energy density *E* (Wh·kg^−1^) was calculated using the equation:(7)$$E=\frac{0.5C_{g}\cdot U^{2}}{3.6},$$
where $C_{g}$ (F·g^−1^) is the specific gravimetric capacitance, *U* (V) is the working potential window.

The corresponding gravimetric power density *P* (W·kg^−1^) was derived from the energy density using the equation:(8)$$P=\frac{3600\cdot E}{t},$$
where t (s) is the total discharge time obtained from the GCD curves.

## 3. Results and Discussion

Figure 1a shows the XRD patterns of the (Mo_2\3_Y_1\3_)_2_AlC MAX phase and the multilayer and delaminated Mo_1.33_CT_x_ MXenes. The (Mo_2\3_Y_1\3_)_2_AlC MAX phase exhibits diffraction peaks at 12.78°, 18.18°, 25.9°, 32.32°, 32.94°, 37.94°, 39.36°, 41.5°, 42.46°, 43.64°, 51.98°, 57.7°, 59.36° and 64.24°, which are assigned to the crystallographic planes (002), (110), (004), (310), (311), (313), (006), (314), (206), (315), (316), (330) and (332), respectively [46]. This confirms the successful formation of the quaternary i-MAX phase, characterized by in-plane chemical ordering of Mo and Y (space group C2/c (#15)). It should also be noted that impurity phases in the form of Y_2_O_3_ and Mo_2_C are present. This is likely due to an oxide film forming on the surface of the metallic yttrium particles. Following the hydrothermal etching of the i-MAX phase in a HCl-LiF mixture, the characteristic diffraction peaks of the i-MAX phase disappear. Additionally, a shift in the (002) diffraction peak from 12.78 to 10.00° was observed. This indicates an increase in the *c*-lattice parameter. This was due to the successful extraction of weakly bound Al atoms from the i-MAX phase structure. Figure 1a shows the XRD pattern of the delaminated MXene film to confirm the successful delamination. As can be seen, only one diffraction peak (002) is present, and the non-basal diffraction peaks have disappeared. This indicates the successful delamination of MXene [47]. The exfoliation and delamination process of the Mo_1.33_CT_x_ MXene is shown schematically in Figure 1b. Figure 1c shows the XPS spectrum of the delaminated MXene. As can be seen, the characteristic peaks of Al and Y are absent due to their complete removal during the hydrothermal etching process. The extraction of Y atoms indicates the successful formation of Mo_1.33_CT_x_ MXene with vacancy ordering [48]. Figure 1d shows an SEM image of multilayer Mo_1.33_CT_x_ MXene particles. Hydrothermal etching resulted in the formation of accordion-like particles consisting of stacks of loosely bound Mo_1.33_CT_x_ MXene nanosheets. Following the intercalation of TMA^+^ ions, the MXenes were successfully delaminated into a single flake, as can be seen in the TEM image (Figure 1e).

Figure 2a–d show the high-resolution XPS spectra of Mo_1.33_CT_x_ in the Mo 3d, C 1s, O 1s, and F 1s regions. The high-resolution XPS spectrum of the Mo 3d region was fitted with three components. The doublet peaks at 229.5 and 232.7 eV were assigned to the Mo–C bond in MXene. The remaining doublet peaks confirmed the presence of the Mo^5+^ (229.9 and 233.1 eV) and Mo^6+^ (232.2 and 235.4 eV) oxide forms. The appearance of these oxide phases is attributed to surface oxidation that occurs during synthesis and subsequent storage in air. The high-resolution XPS spectrum of the C 1s region was fitted with four components. The peak at 283.2 eV corresponds to the C–Mo bond in the MXene structure. The peaks at 284.8, 286.6, and 288.5 eV corresponded to C–C, C–O, and COO bonds, respectively. The high-resolution XPS spectrum of the O 1d region was resolved into four peaks. The peak at 530.6 eV belongs to the molybdenum oxide phases. The peaks at 531.5 and 532.6 eV confirm the presence of oxygen in the form of -O and -OH surface terminations, respectively. The peak at 533.7 eV indicates the presence of adsorbed water. In the F 1s region, the spectrum is approximated by two peaks at 685.8 and 688.9 eV, which correspond to -F surface terminations and AlF_3_ impurities, respectively. Peak-fitting analysis (Appendix A) enabled us to calculate the general formula as Mo_1.25_CO_1.04_(OH)_0.87_F_0.25_Cl_0.42_.

The structure of Mo_1.33_CT_x_ MXene was characterized using Mo K-edge EXAFS. Figure 2e shows the normalized X-ray absorption near-edge structure (XANES) spectra of the Mo_1.33_CT_x_ MXene sample and the reference samples of Mo foil, Mo_2_C, MoO_2_ and MoO_3_. As can be seen, the absorption edges of the reference samples were 20,000, 20,003, 20,007 and 20,009 eV. The absorption edge of Mo in the prepared Mo_1.33_CT_x_ MXene sample was 20,003 eV. To estimate the oxidation state of the materials under study, we created a calibration curve including the Mo foil, Mo_2_C, MoO_2_, and MoO_3_ samples, and assigned the formal oxidation states of 0, +2, +2, +4, and +6, respectively (Figure 2f). This calibration curve enables us to estimate the average oxidation state of molybdenum in Mo_1.33_CT_x_ to be +2.

Figure 3 shows the cyclic voltammetry (CV) curves of Mo_1.33_CT_x_ electrodes in H_2_SO_4_, H_3_PO_4_, LiCl, and KCl electrolytes. The cyclic voltammetry data presented in Figure 3 correspond to electrode coatings with a mass loading of 1.6 mg·cm^−2^. For thicker electrode coatings, the CV curves are shown in Appendix A. As can be seen in Figure 3, in acidic H_2_SO_4_ and H_3_PO_4_ electrolytes, the working potential window ranged from −500 to +200 mV (vs. Ag/AgCl). Further expansion of the potential window beyond 700 mV resulted in electrolyte decomposition. In neutral LiCl and KCl electrolytes, the Mo_1.33_CT_x_ electrodes exhibited stable capacitive behavior within the potential windows of −900 to +100 mV and -800 to 0 mV, respectively. The CV curves of the acidic electrolytes exhibit a pair of redox peaks: an anodic peak at −0.23 V and a cathodic peak at −0.26 V, indicating the contribution of pseudocapacitance to the charge storage mechanism [49]. This is consistent with previous observations of MXenes in aqueous electrolytes [50]. These redox peaks are likely due to the binding of hydronium ions to the oxygen-containing surface terminals of the Mo_1.33_CT_x_ electrode, accompanied by a change in the oxidation state of Mo. Notably, the intensity of the redox peaks in the H_3_PO_4_ electrolyte was lower than in the H_2_SO_4_ electrolyte, despite the similar charge storage mechanism. This is probably due to the conductivity of the electrolytes. The higher conductivity of the H_2_SO_4_ electrolyte provides a higher concentration of intercalated hydronium ions, contributing to a more intense electrode reaction. In neutral LiCl and KCl electrolytes, the CV curves were almost rectangular in shape and lacked redox peaks. This behavior is typical of an electric double-layer capacitor (EDLC) [51]. In these cases, the energy storage process involves only the electrostatic adsorption and desorption of electrolyte ions. Although previous studies have shown that MXene stores charge in LiCl electrolyte through pseudocapacitive lithium-ion intercalation—leading to a higher specific capacitance than electric double-layer capacitance [52]—no distinct redox peaks were detected on the CV curves in either LiCl or KCl electrolytes. This suggests that the charge storage likely follows a hybrid mechanism, combining both electric double-layer capacitance and pseudocapacitance.

Appendix A shows the CV curves of Mo_1.33_CT_x_ electrodes with different mass loadings of active material in H_2_SO_4_, H_3_PO_4_, LiCl, and KCl electrolytes. As can be seen, the electrode thickness has a significant effect on the CV profiles at high scan rates. The CV curves of thin electrodes did not deviate significantly with increasing scan rates. However, increasing the electrode thickness resulted in substantial distortion of the CV curves. It is well-known that electrode thickness significantly affects charge transfer processes and energy storage performance. Increasing the thickness greatly hinders the diffusion of cations, particularly at high scan rates. Figure 4a–c show how the specific gravimetric capacitance depends on the scan rate for electrodes with mass loadings of 1.6, 3.5, and 5.3 mg∙cm^−2^. At 2 mV∙s^−1^, the specific capacitances of the thin electrodes (1.6 mg∙cm^−2^) were 352, 287, 172, and 107 F∙g^−1^ in H_2_SO_4_, H_3_PO_4_, LiCl, and KCl electrolytes, respectively. Increasing the scan rate to 100 mV∙s^−1^ resulted in a decrease in the specific capacitance values to 313, 195, 103, and 59 F∙g^−1^ in H_2_SO_4_, H_3_PO_4_, LiCl, and KCl electrolytes, respectively. In the H_2_SO_4_ electrolyte, the capacitance retention was high, with only a 12% decrease. This is attributed to the facile transport of small electrolyte ions (protons) within the electrode structure [53,54]. Despite the similar mechanism of pseudocapacitive charge accumulation, a significant decrease in specific capacitance was observed in the H_3_PO_4_ electrolyte at high scan rates. This is due to the lower conductivity of the H_3_PO_4_ electrolyte. With an increasing scan rate of 100 mV∙s^−1^, the specific capacitance of the LiCl electrolyte decreased by 40%. Considering the large radius of solvated Li^+^ ions, this is a good result. It should be noted that the specific gravimetric capacitance of Mo_1.33_CT_x_ electrodes in an H_2_SO_4_ electrolyte is comparable to that of Ti_3_C_2_T_x_ MXene. Furthermore, Mo_1.33_CT_x_ MXene exhibits the highest specific capacitance in the LiCl electrolyte compared to other known MXenes. This enhanced performance is likely due to the presence of ordered vacancies, which can act as active sites for redox reactions and facilitate ion intercalation, thereby boosting pseudocapacitance. In a KCl electrolyte, however, Mo_1.33_CT_x_ electrodes demonstrated the lowest performance. The significant differences in specific capacitance between the LiCl and KCl electrolytes can be explained by differences in charge propagation. In MXene electrodes, the rapid surface diffusion of solvated K^+^ ions appears to limit charge propagation, whereas the slower but deeper intercalation of solvated Li^+^ ions enables higher capacitance [55]. As the mass loading of the electrodes increased to 3.5 mg∙cm^−2^, the specific capacitances decreased slightly at low scan rates. At a scan rate of 2 mV∙s^−1^, the specific capacitances of the electrodes were 349, 260, 162, and 98 F∙g^−1^ in H_2_SO_4_, H_3_PO_4_, LiCl, and KCl electrolytes, respectively. However, at 100 mV∙s^−1^, the specific capacitance values decreased to 187, 91, 57, and 36 F∙g^−1^, respectively. Increasing the electrode thickness further to 5.3 mg∙cm^−2^ resulted in an even more significant decrease in specific capacitance with increasing scan rate. Additionally, at low scan rates, the specific capacitances decreased in all electrolytes. At 2 mV∙s^−1^, the specific capacitances were 294, 248, 158, and 87 F∙g^−1^ in H_2_SO_4_, H_3_PO_4_, LiCl, and KCl electrolytes, respectively. At 100 mV∙s^−1^, the specific capacitances decreased to 111, 26, 23, and 7 F∙g^−1^ in H_2_SO_4_, H_3_PO_4_, LiCl, and KCl electrolytes, respectively. For this thickest electrode, ion transport within the bulk was significantly hindered, limiting access to the redox-active sites on the Mo_1.33_CT_x_ surface. The decrease in specific capacitance at low scan rates indicates that not the entire electrode volume was involved in charge storage due to the diffusion barrier for electrolyte ions. Nevertheless, the specific capacitance remained high despite the electrode’s very high mass loading at low scan rates. This was attributed to the influence of carbon nanotubes on the electrode structure. It is important to note that CNTs do not contribute directly to energy storage, as they are electrochemically inactive within the potential window in which Mo_1.33_CT_x_ is active. Nevertheless, CNTs act as a conductive separator, disrupting the usual horizontal arrangement of Mo_1.33_CT_x_ nanosheets [56]. This maximizes the accessibility of electrolyte ions to the entire active surface area of the electrode.

Figure 4d–f show the dependence of the logarithm of the peak current CV on the logarithm of the scan rate for electrodes of different thicknesses. To explain the kinetics of charge accumulation in the electrodes, the parameter *b* in Equation (3) was determined from the slope of the fitted lines. As can be seen in Figure 4d–f, the *b* parameter for the electrodes with a mass loading of 1.6 mg∙cm^−2^ was 1.00, 0.94, 0.88, and 0.88 for the H_2_SO_4_, H_3_PO_4_, LiCl, and KCl electrolytes, respectively. Values of *b* close to 1 imply a surface-capacitive charge accumulation mechanism, which is dominant in the acidic electrolytes. For the LiCl and KCl electrolytes, the *b* values between 0.5 and 1 suggest a hybrid energy storage mechanism, combining capacitive and diffusion-controlled processes. As the mass loading of the electrodes increased, the *b* values gradually decreased in all cases. This confirms the growing role of diffusion processes in charge accumulation. Additionally, a transition to a mechanism controlled entirely by diffusion processes was observed for electrodes with a mass loading of 5.3 mg∙cm^−2^ in a KCl electrolyte.

To further analyze the charge accumulation mechanism, the contributions of surface-controlled and diffusion-controlled processes were estimated using Equation (6). Appendix A shows the CV curves of electrodes of different thicknesses at a scan rate of 5 mV∙s^−1^. The shaded area of the CV curves indicates the contribution of surface-capacitive processes. The numerical values showing the contribution of surface-capacitive processes are presented in Figure 4g–i. For thin electrodes, the contribution of surface-capacitive processes was 88.9%, 75.7%, 71.2% and 66.8% in H_2_SO_4_, H_3_PO_4_, LiCl, and KCl, respectively, at 5 mV∙s^−1^. As the electrode thickness increased, the contribution of surface-capacitive processes gradually decreased. Notably, in H_2_SO_4_, the contribution of surface-capacitive processes remained predominant even at high electrode material mass loadings. This indicates a charge accumulation mechanism associated with controlled surface oxidation-reduction reactions. In the H_3_PO_4_ electrolyte, the prevalence of diffusion processes increased with increasing electrode thickness. This is due to the low conductivity of the electrolyte, which creates a barrier to the transport of electrolyte ions into the electrode volume. In the LiCl electrolyte, the contribution of surface-capacitive processes was about 70% at a specific electrode loading of 1.6 to 3.5 mg·cm^−2^. This behavior is probably caused by the presence of ordered vacancies that facilitate the intercalation of Li^+^ ions. Compared to the LiCl electrolyte, the predominance of surface-capacitive processes was less obvious in the KCl electrolyte, especially at high electrode material loading. The significant difference in charge storage kinetics between KCl and LiCl electrolytes can be attributed to the distinct behavior of their ions. Despite difficulties with desolvation, the small lithium ions can penetrate deeply into the material’s interlayer spaces. This allows the material’s entire internal surface area to be utilized for charge storage and enables pseudocapacitive reactions. In contrast, the larger potassium ions remain confined to the outer surface, which severely limits the accessible area and, consequently, the overall capacity. Thus, the electrode film thickness plays a critical role in determining the energy storage efficiency at different scan rates. At a low specific mass loading (1.6 mg∙cm^−2^), the electrolyte ions can easily access a large portion of the electrode’s specific surface area, even at high scan rates. As the loading increases to 3.5 mg·cm^−2^, ion access to the entire specific surface area becomes limited at high scan rates. Furthermore, for thick electrodes (5.3 mg∙cm^−2^), this limitation occurs even at low scan rates. Based on these results, it can be concluded that a mass loading of approximately 1.6 mg·cm^−2^ is optimal for high-rate supercapacitors. Meanwhile, for applications not requiring high charge–discharge rates, a loading of up to 3.5 mg·cm^−2^ is acceptable. The use of thicker coatings is not recommended, as it leads to inefficient active material utilization.

Figure 5a–d show the galvanostatic charge–discharge (GCD) curves at various current densities. As can be seen, the shape of the curves varies depending on the type of electrolyte. In acidic H_2_SO_4_ and H_3_PO_4_ electrolytes, the profiles deviate significantly from ideal straight triangular lines due to the predominance of pseudocapacitance. The charge and discharge profiles are symmetrical, indicating high reversibility of the redox reactions on the surface of Mo_1.33_CT_x_ nanosheets. The GCD curves for LiCl and KCl electrolytes closely resemble the ideal triangular profile since charge accumulation occurs through the rapid intercalation and deintercalation of cations, resulting in the formation of an electric double layer. Small deviations of the discharge curves from the ideal triangular profile can be attributed to the irreversible intercalation of Li+ and K+ ions. As can be seen, the discharge time in the H_2_SO_4_ electrolyte was longer than in the H_3_PO_4_ electrolyte at the same current densities. Similarly, the discharge time in the LiCl electrolyte was longer than in the KCl electrolyte. These results confirm that the specific capacitance of these electrolytes is higher, which is consistent with the CV data. The specific capacitances were determined from the discharge curve profiles at different current densities, as shown in Figure 5e,f. The specific capacitances in the H_2_SO_4_ electrolyte were 300, 221, and 214 F∙g^−1^ at current densities of 2.5, 5, and 10 A∙g^−1^, respectively. At the same current densities, the specific capacitances in the H_3_PO_4_ electrolyte were 250, 178, and 164 F∙g^−1^. As can be seen, the electrodes demonstrated stable capacitive behavior in acidic electrolytes even at high current densities. In LiCl and KCl electrolytes, the maximum current density was 5 A∙g^−1^, since the charge storage efficiency dropped sharply at higher current densities. In the LiCl electrolyte, the specific capacitance was 172, 169, and 150 F∙g^−1^ at current densities of 1, 2, and 5 A∙g^−1^, respectively. In the KCl electrolyte, the specific capacitance was significantly lower. At current densities of 1, 2, and 5 A∙g^−1^, the specific capacitance was 127, 107, and 78 F∙g^−1^, respectively. Energy and power densities were calculated using Equations (7) and (8). The Mo_1.33_CT_x_ electrodes exhibited gravimetric energy densities of 20.4 and 17.0 Wh·kg^−1^ in H_2_SO_4_ and H_3_PO_4_ electrolytes, respectively, at a current density of 2.5 A·g^−1^, with corresponding power densities of 875 W·kg^−1^ in both cases. When tested in LiCl and KCl electrolytes at 1 A·g^−1^, the energy densities were 38.1 and 11.3 Wh·kg^−1^, respectively, while the corresponding power densities reached 500 and 400 W·kg^−1^.

In conclusion, Mo_1.33_CT_x_ MXene electrodes not only exhibited outstanding energy storage properties, but also high-capacity stability under long-term cycling. As illustrated in Appendix A, the specific capacitance decreased by just 10% after 10,000 charge–discharge cycles in H_2_SO_4_ electrolyte at a current density of 10 A∙g^−1^. For a comparison of the electrochemical performance, the specific capacitance values of the MXene-based electrodes are listed in Table 1. The data show that Mo_1.33_CT_x_ exhibits superior energy storage performance in LiCl electrolyte compared to Ti_3_C_2_T_x_. At the same time, the specific capacitance of Mo_1.33_CT_x_ in acidic H_2_SO_4_ and H_3_PO_4_ electrolytes is comparable to that of Ti_3_C_2_T_x_. This highlights the potential of Mo_1.33_CT_x_ MXene as a material for developing reliable and efficient energy storage systems. The results obtained open up new possibilities for the further development of MXene-based electrode materials in modern electrochemical devices.

## 4. Conclusions

In this study, we successfully synthesized a two-dimensional Mo_1.33_CT_x_ MXene with ordered vacancies by selectively extracting Al and Y atoms from the quaternary MAX phase (Mo_2/3_Y_1/3_)_2_AlC under hydrothermal conditions. Comprehensive characterization of the resulting material using XRD, SEM, TEM, XANES, and XPS methods confirmed the formation of single-layer MXene flakes with the composition Mo_1.25_CO_1.04_(OH)_0.87_F_0.25_Cl_0.42_, in which the predominant oxidation state of molybdenum is +2.

A scalable doctor-blade coating method was employed to fabricate supercapacitor electrodes using a Mo_1.33_CT_x_-CNTs–PVDF slurry. By systematically optimizing the mass loading (1.6, 3.5, and 5.3 mg∙cm^−2^), we established a direct correlation between the electrochemical performance and the electrode coating thickness. Electrodes with low mass loading (1.6 mg∙cm^−2^) exhibited outstanding specific capacitance values—352, 287, 172, and 107 F∙g^−1^ in H_2_SO_4_, H_3_PO_4_, LiCl, and KCl electrolytes, respectively, at a scan rate of 2 mV∙s^−1^- along with excellent charge transport kinetics, even at elevated scan rates. These characteristics make them highly suitable for high-power and fast-response supercapacitor applications. Electrodes with intermediate mass loading (3.5 mg∙cm^−2^) showed a moderate decline in specific capacitance (349, 260, 162, and 98 F∙g^−1^ in H_2_SO_4_, H_3_PO_4_, LiCl, and KCl, respectively), primarily due to increased ion diffusion resistance. Nevertheless, they retained high energy storage capacity at low scan rates, which is advantageous for miniaturized devices operating under less demanding power conditions. In contrast, heavily loaded electrodes (5.3 mg∙cm^−2^) demonstrated significantly lower electrochemical performance, attributable to pronounced diffusion limitations within the thick active layer. These findings highlight the critical importance of optimizing electrode thickness and mass loading to achieve a balance between energy and power density in a supercapacitor. Using an H_2_SO_4_ electrolyte enables the highest specific capacitance to be achieved due to its high proton conductivity. However, despite a slight decrease in energy efficiency, the H_3_PO_4_ electrolyte can also be considered for developing high-performance supercapacitors that require increased chemical stability. The electrodes showed significantly higher performance in the LiCl electrolyte than in the KCl electrolyte. This was due to the smaller ionic radius of Li+, which facilitates intercalation and increases charge mobility. The results obtained demonstrate the potential of blade-coated Mo_1.33_CT_x_-CNTs electrodes for the scalable fabrication of high-performance supercapacitors with adjustable characteristics depending on the target application.

## Figures and Tables

**Figure 1 nanomaterials-15-01593-f001:**
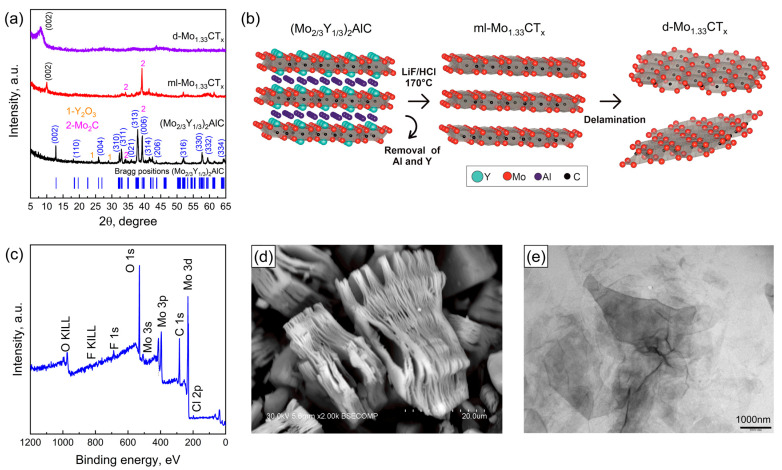
XRD patterns of the (Mo_2\3_Y_1\3_)_2_AlC MAX phase, the multilayer ml-Mo_1.33_CT_x_ MXene, and the delaminated d-Mo_1.33_CT_x_ MXene film (**a**). A schematic representation shows the transformation of (Mo_2\3_Y_1\3_)_2_AlC into Mo_1.33_CT_x_ MXene nanosheets (**b**). XPS survey spectrum of Mo_1.33_CT_x_ MXene (**c**). SEM images of ml-Mo_1.33_CT_x_ (**d**) and TEM images of Mo_1.33_CT_x_ nanosheets (**e**).

**Figure 2 nanomaterials-15-01593-f002:**
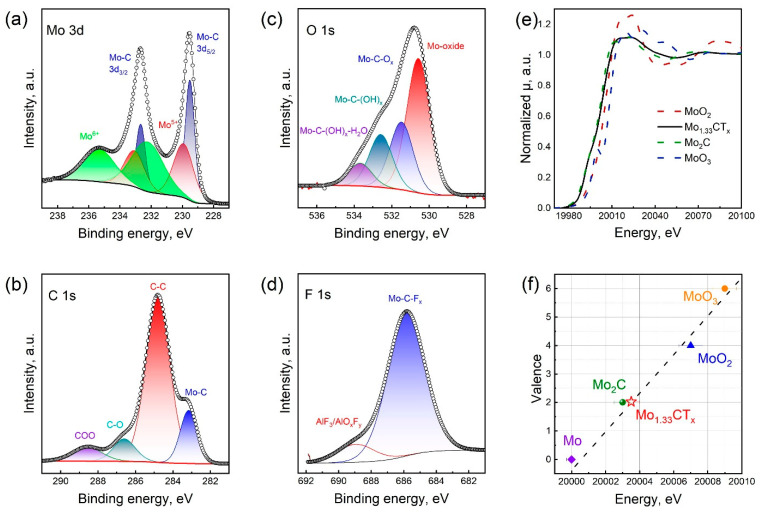
High resolution XPS spectra of Mo_1.33_CT_x_ MXene in the Mo 3d (**a**), C 1s (**b**), O 1s (**c**) and F 1s (**d**) regions. Normalized XANES spectra of Mo K-edge for MoO_2_, Mo_2_C, MoO_3,_ and Mo_1.33_CT_x_ MXene (**e**). Fitted chemical valences for the Mo atom in Mo_1.33_CT_x_ MXene (**f**).

**Figure 3 nanomaterials-15-01593-f003:**
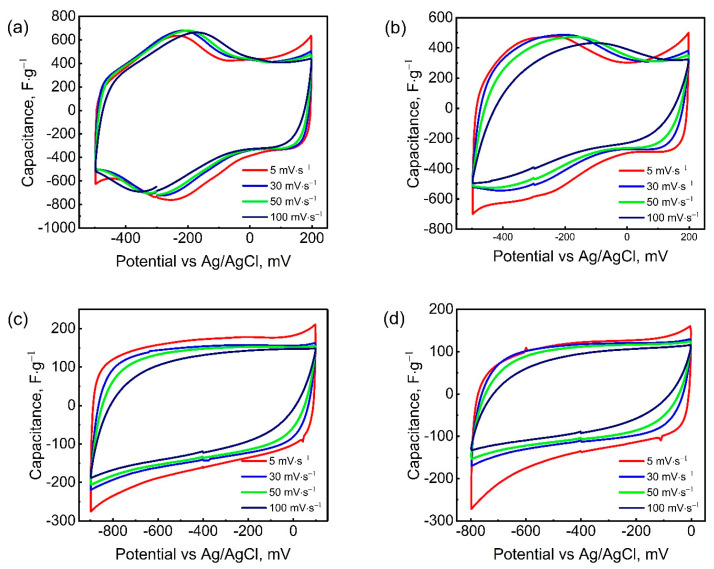
CV curves of Mo_1.33_CT_x_ electrode at different scan rates in H_2_SO_4_ (**a**), H_3_PO_4_ (**b**), LiCl (**c**), and KCl (**d**) electrolytes.

**Figure 4 nanomaterials-15-01593-f004:**
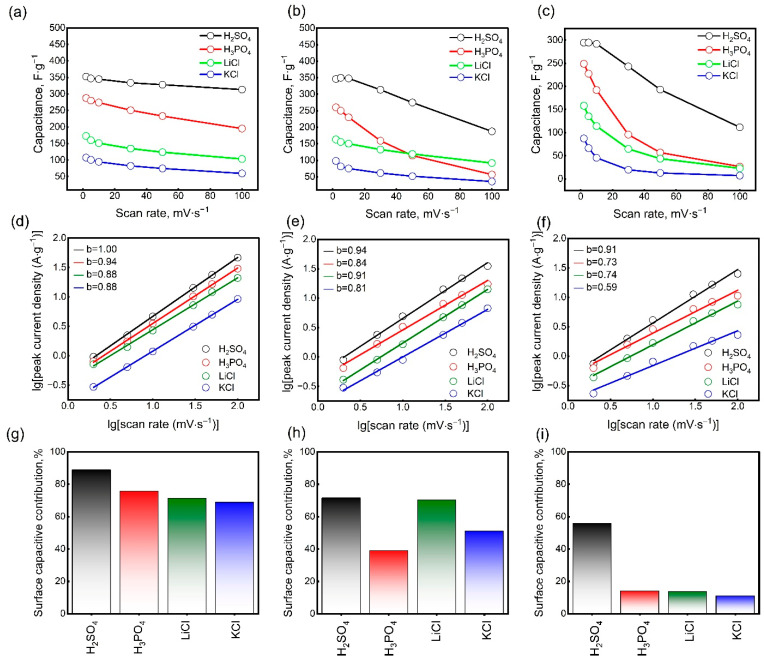
The specific gravimetric capacitances of Mo_1.33_CT_x_ electrodes with mass loadings of 1.6 (**a**), 3.5 (**b**), and 5.3 (**c**) mg∙cm^−2^. Logarithm of peak currents versus logarithm of the scan rate for Mo_1.33_CT_x_ electrodes with mass loadings of 1.6 (**d**), 3.5 (**e**), and 5.3 (**f**) mg∙cm^−2^ for the determination of *b*-values. Contribution of surface-controlled processes to charge storage for Mo_1.33_CT_x_ electrodes with mass loadings of 1.6 (**g**), 3.5 (**h**), and 5.3 (**i**) mg∙cm^−2^.

**Figure 5 nanomaterials-15-01593-f005:**
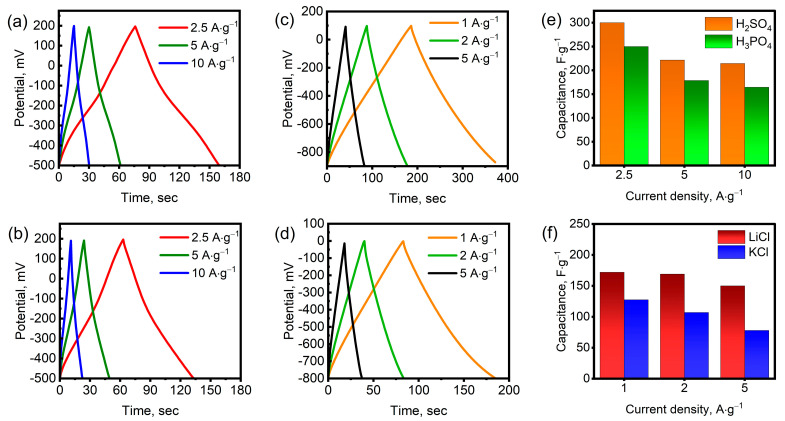
GCD curves for Mo_1.33_CT_x_ in H_2_SO_4_ (**a**), H_3_PO_4_ (**b**), LiCl (**c**), and KCl (**d**) electrolytes. Specific capacitance at various current densities in different electrolytes (H_2_SO_4_ and H_3_PO_4_ (**e**), LiCl and KCl (**f**)).

**Table 1 nanomaterials-15-01593-t001:** Electrochemical performance of MXene-based supercapacitor electrodes.

Electrode Material	Scan Rate, mV·s^−1^	Electrolyte	Capacitance, F·g^−1^	Ref.
Ti_3_C_2_T_x_	5	5 M LiCl	120	[57]
	10	1 M H_2_SO_4_	340	[58]
	2	9 M H_3_PO_4_	280	[59]
V_2_CT_x_	2	1 M H_2_SO_4_	487	[60]
Mo_4_VC_4_T_x_	2	5 M LiCl	66	[61]
		3 M H_2_SO_4_	219	
Mo_1.87_CT_x_	2	1 M H_2_SO_4_	304	[62]
Mo_2_Ti_2_C_3_T_x_	3	1 M H_2_SO_4_	110	[63]
Mo_1.33_CT_x_	2	2 M H_2_SO_4_	352	This work
		1 M H_3_PO_4_	287	
		5 M LiCl	172	

## Data Availability

The raw data supporting the conclusions of this article will be made available by the authors on request.

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
