# Peer review of "Pseudocapacitive Behavior of Blade-Coated Mo_1.33_CT_x_ i-MXene Electrodes in Aqueous Electrolytes"

_nanomaterials, 2025, doi:10.3390/nano15201593_

Round 1
Reviewer 1 Report
Comments and Suggestions for Authors
I have reviewed the manuscript entitled “Pseudocapacitive Behavior of Blade-Coated Mo1.33CTx i-MXene Electrodes in Aqueous Electrolytes” by N. Gorshkov et al, submitted to Nanomaterials/MDPI. In the submitted manuscript, the authors investigate blade-coated molybdenum carbide (Mo₁.₃₃CTₓ) MXene electrodes in aqueous electrolytes, demonstrating their pseudocapacitive behavior. This behavior arises from rapid ion and electron transport facilitated by redox reactions, contributing to high energy storage performance. The electrochemical characteristics—such as voltage window and specific capacitance—are shown to be strongly dependent on the type of electrolyte used, whether hydronium-based or cation-based (e.g., LiCl or KCl). Molybdenum-based MXenes, particularly in LiCl electrolytes, exhibit wide voltage windows and high volumetric capacitance, highlighting their potential for high-performance supercapacitor applications. Furthermore, increasing the electrode mass loading from 1.6 to 3.5 mg/cm² significantly influences the specific capacitance. While the manuscript presents promising findings, certain sections lack clarity and could benefit from improved organization and language to enhance readability and accessibility for the target audience. My specific points are given below.
- The different types of supercapacitors and their mechanisms could be briefly described in the introduction section at the end of the first paragraph.
- The importance of blade coating should be emphasized in the introduction.
- The pH values and molar concentrations of the electrolytes used can be detailed in the methodology section.
- How were the energy and power densities calculated?
- In Section 3, line 232, will the TMA+ ions participate in the electrochemical reaction?
- How do the thin and thick electrodes affect the energy storage properties?
- On page 8, lines 280 and 283, “redox peaks” and “intensity of redox peaks” provide the appropriate potential values and the proposed mechanism for the redox peaks.
- The electrolytes H3PO4 and H2SO4, which are similar and reported in the literature for aqueous supercapacitors, must be compared for their storage mechanisms, such as doi.org/10.1002/cplu.202500342 and doi.org/10.1002/asia.202400622.
- Line 288, usually an electric double-layer capacitor (EDLC) mechanism involves ion adsorption and desorption, but here, intercalation of Li+ ions is proposed (see line 381). Justify.
- In Figure 5f, is it 1, 2, and 5 A/g or 1, 2, and 10 A/g? Please check.
- What is the practical application of these proposed electrodes and electrolytes, considering the device study has not been mentioned in the work?
Author Response
- The different types of supercapacitors and their mechanisms could be briefly described in the introduction section at the end of the first paragraph.
Response:
Thank you for your valuable feedback. We have added a description of EDLC and pseudocapacitors to the introduction.
- The importance of blade coating should be emphasized in the introduction.
Response:
Thank you for your feedback. We have addressed this point in the revised version of the manuscript.
- The pH values and molar concentrations of the electrolytes used can be detailed in the methodology section.
Response:
The pH values of the electrolytes used have been added to the "Materials and Methods" section.
- How were the energy and power densities calculated?
Response:
Thank you for your comment. The methodology for calculating the specific energy and power densities has been added to the manuscript.
- In Section 3, line 232, will the TMA+ ions participate in the electrochemical reaction?
Response:
Тetramethylammonium ions are not electrochemically active in the MXene electrode system. They should be viewed as a tool for structural modification during the electrode preparation stage. They act as "spacers" that physically expand the MXene interlayer galleries, thereby significantly increasing the available surface area and facilitating electrolyte ion transport.
- How do the thin and thick electrodes affect the energy storage properties?
Response:
Increasing the thickness of the electrode coating reduces its specific surface area accessible to electrolyte ions. We have added a discussion of this point to the revised manuscript.
- On page 8, lines 280 and 283, “redox peaks” and “intensity of redox peaks” provide the appropriate potential values and the proposed mechanism for the redox peaks.
The electrolytes H3PO4 and H2SO4, which are similar and reported in the literature for aqueous supercapacitors, must be compared for their storage mechanisms, such as doi.org/10.1002/cplu.202500342 and doi.org/10.1002/asia.202400622.
Response:
Thank you for your recommendations. We have taken the suggested literature into consideration.
- Line 288, usually an electric double-layer capacitor (EDLC) mechanism involves ion adsorption and desorption, but here, intercalation of Li+ ions is proposed (see line 381). Justify.
Response:
Thank you for your valuable comment. We postulate that the energy storage proceeds via a hybrid mechanism that combines the electric double-layer capacitor (EDLC) effect with pseudocapacitive intercalation. We have added corresponding clarifications to the revised version of the manuscript.
- In Figure 5f, is it 1, 2, and 5 A/g or 1, 2, and 10 A/g? Please check.
Response:
Thank you for pointing this out. It was a technical error, which has now been corrected in the figure 5с-d.
- What is the practical application of these proposed electrodes and electrolytes, considering the device study has not been mentioned in the work?
- Response:
The practical application of the studied materials is expected in symmetric supercapacitors with aqueous electrolytes.
Reviewer 2 Report
Comments and Suggestions for Authors
In this manuscript, the authors have demonstrated the fabrication of Mo1.33CTx/carbon nanotubes (CNTs) composite to use as active electrodes for high performance supercapacitors. They have studied their electrochemical performance in different aqueous electrolyte. The manuscript is well prepared and explained the major experimental outcomes. Therefore, I am pleased to accept this manuscript after minor revision. My specific comments are given below.
- They have introduced CNT to increase the interlayer distance of the MXenes. However, they need to explain how CNT can do that and give some experimental proof.
- Electrochemical performance of the Mo1.33CTx/carbon nanotubes (CNTs) was studied in the different electrolytes. However, they need to provide the performance of the Mo1.33CTx MXenes to compare how effectively CNT increase the interlayer distance and effects the electrochemical performance.
- Contribution of surface-controlled processes in charge storage of Mo1.33CTx electrodes varies with different mass loading. However, the trend is not following the same in different mass loading. Any specific reason for that.
- Energy density and power density are very important parameters to evaluate the electrochemical performance of supercapacitor. Authors need to provide such information in the revised manuscript. Also provide a comparison table with recent electrochemical performance of others MXenes composite.
Author Response
- They have introduced CNT to increase the interlayer distance of the MXenes. However, they need to explain how CNT can do that and give some experimental proof.
Response:
Thank you for this comment. The effect of CNTs on the architecture of MXene electrodes is indeed a well-known phenomenon, and relevant references have been provided in the introduction. To further confirm the increase in the interlayer spacing of the MXene nanosheets, we have included the XRD pattern of the electrodes in the Supporting Information (Fig. S1), along with a corresponding discussion of the results.
- Electrochemical performance of the Mo1.33CTx/carbon nanotubes (CNTs) was studied in the different electrolytes. However, they need to provide the performance of the Mo1.33CTx MXenes to compare how effectively CNT increase the interlayer distance and effects the electrochemical performance.
Response:
We appreciate the reviewer's remark. As outlined in the introduction, the effect of carbon nanotubes (CNTs) on various MXenes has been documented in prior research. We concur that illustrating the specific impact of CNTs on electrochemical performance would be valuable. Nevertheless, such a direct comparison is complicated by our experimental methodology, specifically the use of PVDF binder in electrode fabrication. The incorporation of this insulating binder diminishes the conductivity of the electrode film. Consequently, the electrochemical data for Mo1.33CTx /PVDF without CNTs could be anomalous and not representative of the material's intrinsic properties, potentially leading to incorrect conclusions. Therefore, in the context of this study, CNTs were employed with a dual function: firstly, to separate the MXene nanosheets, thereby facilitating ion transport, and secondly, to serve as a conductive additive to compensate for the resistivity introduced by the PVDF binder.
- Contribution of surface-controlled processes in charge storage of Mo1.33CTx electrodes varies with different mass loading. However, the trend is not following the same in different mass loading. Any specific reason for that.
Response:
Thank you for your comment. You are correct that an increase in the electrode coating thickness limits the electrolyte ions' access to the specific surface area of the electrode. We have added clarifications on this point to the manuscript.
- Energy density and power density are very important parameters to evaluate the electrochemical performance of supercapacitor. Authors need to provide such information in the revised manuscript. Also provide a comparison table with recent electrochemical performance of others MXenes composite.
Response:
Thank you for your valuable feedback. We have incorporated the requested parameters into the revised manuscript. Additionally, a table has been included to enable a comprehensive comparison of the electrochemical performance with other MXenes.
Reviewer 3 Report
Comments and Suggestions for Authors
The authors studied the pseudocapacitive behavior of a novel Mo-based MXene. In fact, the pseudocapacitance of other types of MXenes has already been widely investigated, but this work does not reveal significant differences from previous studies. Therefore, the authors must add additional characterizations to make the research more complete, and at the same time provide detailed comparisons and analyses with previous studies to clearly demonstrate the innovation of this work. Specific comments are as follows:
- The authors state: “However, the high defectivity and tendency to aggregate of nanosheets hinders the large-scale fabrication of …”. This seems to be the key scientific issue in this field. How does this work address the problem of “high defectivity and tendency to aggregate of nanosheets”?
- Compared with different types of MXenes, especially Ti-based MXenes, what are the differences in performance? What are the advantages and disadvantages? This directly determines the value of the work. A comparison is needed, with more insightful analysis and new knowledge.
- How is the long-cycle stability of the material?
- What about the reaction kinetics? What is the interfacial resistance? Please refer to Fig. 6 in the paper Compressible and Lightweight MXene/Carbon Nanofiber Aerogel with “Layer-Strut” Bracing Microscopic Architecture for Efficient Energy Storage. Please provide EIS spectra and analysis.
- The XPS analysis needs to be carefully checked, especially for Mo. The differences in full width at half maximum are too large, and the peaks overlap, which does not conform to the standard for peak fitting.
- The authors claim that CNTs are inert within the electrochemical window of MXene and do not store energy. What is the source of this statement? Please perform electrochemical testing within this range and provide evidence.
- Since the pseudocapacitance of MXene is the core of this work, the authors should summarize and review the progress of MXene pseudocapacitance. Many important papers in this area have been largely ignored, for example: Ultrahigh energy and power densities of d-MXene-based symmetric supercapacitors, Nanomaterials, 2022, 12(19): 3294; MXene based nanocomposite films; Unlocking Novel Functionality: Pseudocapacitive Sensing in MXene-Based Flexible Supercapacitors.
- The supporting information shows that the carbon content is much higher than its stoichiometric ratio. Why?
- Compared with other types of MXenes, how is the energy storage performance of Mo-based MXene? Please provide comparisons. The following are closely related papers, and the authors should also look for more relevant literature: Bidirectionally aligned MXene hybrid aerogels assembled with MXene nanosheets and microgels for supercapacitors; NH3-Induced In Situ Etching Strategy Derived 3D-Interconnected Porous MXene/Carbon Dots Films for High Performance Flexible Supercapacitors.
- The capacity differs significantly in different electrolytes. The underlying mechanism needs to be specifically analyzed.
- Carefully check the LiCl data in Fig. 4h. This dataset seems identical to that in Fig. 4g. What is the reason for this anomaly?
Author Response
- The authors state: “However, the high defectivity and tendency to aggregate of nanosheets hinders the large-scale fabrication of …”. This seems to be the key scientific issue in this field. How does this work address the problem of “high defectivity and tendency to aggregate of nanosheets”?
Response:
Thank you for your comment. While MXene electrodes are typically fabricated as free-standing, binder-free films, this approach is problematic for Mo₁.₃₃C-based materials. To create robust Mo₁.₃₃C electrodes, the use of a binder such as PVDF is necessary. It should be noted that the issue of material defectivity is not addressed in this study; however, PVDF is employed here specifically to produce durable electrodes. Furthermore, carbon nanotubes (CNTs) are incorporated into the electrode composition to prevent agglomeration and compensate for electrical conductivity. This combination ultimately enables the fabrication of robust and high-performance electrodes. We have added clarifications on this matter in the revised manuscript.
- Compared with different types of MXenes, especially Ti-based MXenes, what are the differences in performance? What are the advantages and disadvantages? This directly determines the value of the work. A comparison is needed, with more insightful analysis and new knowledge.
Response:
Thank you for your comment. In response, we have included a comparative table of MXene-based electrodes. Furthermore, the suggested literature sources have been added to the reference list.
- How is the long-cycle stability of the material?
Response:
The cycling stability data are summarized in Fig. S3, and corresponding comments are provided in the main text. To investigate the stability under the most demanding conditions, we used the most aggressive electrolyte, H₂SO₄. The specific capacitance retained 90% of its initial value after 10,000 charge-discharge cycles in H₂SO₄ at a current density of 10 A·g⁻¹.
- What about the reaction kinetics? What is the interfacial resistance? Please refer to Fig. 6 in the paper Compressible and Lightweight MXene/Carbon Nanofiber Aerogel with “Layer-Strut” Bracing Microscopic Architecture for Efficient Energy Storage. Please provide EIS spectra and analysis.
Response:
A truly important aspect of electrochemical energy storage is the kinetics of charge transfer across the electrode-electrolyte interface, as well as, when present, the kinetics of the electrode reaction or the diffusion of charge carriers within the electrode. We limited our study to voltammetric investigations, which are presented in detail in the supplementary file and discussed in the manuscript. They are no less informative than impedance studies. The description has been slightly expanded.
- The XPS analysis needs to be carefully checked, especially for Mo. The differences in full width at half maximum are too large, and the peaks overlap, which does not conform to the standard for peak fitting.
Response:
While we acknowledge the reviewer's comment, a more rigorous quantification of Mo⁶⁺ is not considered feasible.
- The authors claim that CNTs are inert within the electrochemical window of MXene and do not store energy. What is the source of this statement? Please perform electrochemical testing within this range and provide evidence.
Response:
According to the literature, CNTs exhibit electrochemical activity at potentials more positive than those of the Ag/AgCl reference electrode. Furthermore, the small proportion of CNTs used in the composite allows their contribution to the overall electrochemical energy storage to be considered negligible.
- Since the pseudocapacitance of MXene is the core of this work, the authors should summarize and review the progress of MXene pseudocapacitance. Many important papers in this area have been largely ignored, for example: Ultrahigh energy and power densities of d-MXene-based symmetric supercapacitors, Nanomaterials, 2022, 12(19): 3294; MXene based nanocomposite films; Unlocking Novel Functionality: Pseudocapacitive Sensing in MXene-Based Flexible Supercapacitors.
Response:
Thank you for the helpful references. They have been added to the bibliography in the revised version of the manuscript.
- The supporting information shows that the carbon content is much higher than its stoichiometric ratio. Why?
Response:
Analysis of the XPS survey spectrum indeed shows a high carbon content. This is a typical phenomenon for MXenes. For example, similar results were obtained previously: DOI: 10.1002/adfm.201505328, https://doi.org/10.1016/j.carbon.2021.03.062. This is likely due to the fact that the penetration depth of XPS radiation does not exceed 10 nm. At the same time, thin carbonaceous layers form on the surface of MXene films, which is associated with contamination caused by intercalation and interaction with the ambient environment. This leads to the appearance of an intense C 1s peak. Thus, this does not mean that the MXene content in the sample is significantly overestimated, but rather indicates that thin carbonaceous layers are formed on the surface of the powders.
- Compared with other types of MXenes, how is the energy storage performance of Mo-based MXene? Please provide comparisons. The following are closely related papers, and the authors should also look for more relevant literature: Bidirectionally aligned MXene hybrid aerogels assembled with MXene nanosheets and microgels for supercapacitors; NH3-Induced In Situ Etching Strategy Derived 3D-Interconnected Porous MXene/Carbon Dots Films for High Performance Flexible Supercapacitors.
Response:
Thank you for your comment. In response, we have included a comparative table of MXene-based electrodes. Furthermore, the suggested literature sources have been added to the reference list.
- The capacity differs significantly in different electrolytes. The underlying mechanism needs to be specifically analyzed.
Response:
Thank you for your remark. We have included a commentary comparing the charge storage mechanisms across various electrolytes.
- Carefully check the LiCl data in Fig. 4h. This dataset seems identical to that in Fig. 4g. What is the reason for this anomaly?
Response:
We have completed the re-examination of the data, which, among other things, contributed to the delay in responding to the review. We can confidently state that the shape of the CV curves remains unchanged with variations in electrode thickness, and the coincidence in values is due to the identical electrode area. This is explained by the dominant contribution of the surface-capacitance mechanism.
Round 2
Reviewer 1 Report
Comments and Suggestions for Authors
The revised version is suitable for publication.
Reviewer 3 Report
Comments and Suggestions for Authors/